# GDF2 and BMP10 coordinate liver cellular crosstalk to maintain liver health

Dianyuan Zhao[1]*[†], Ziwei Huang[1,2][†], Xiaoyu Li[1,2][†], Huan Wang[1][†], Qingwei Hou[1,3], Yuyao Wang[1,3], Fang Yan[1], Wenting Yang[1], Di Liu[1], Shaoqiong Yi[1], Chunguang Han[1], Yanan Hao[1], Li Tang[1,2,3]*

[1]State Key Laboratory of Medical Proteomics, Beijing Proteome Research Center, National Center for Protein Sciences, Beijing Institute of Lifeomics, Beijing, China; [2]Department of Immunology, School of Basic Medical Sciences, Anhui Medical University, Hefei, China; [3]School of Basic Medicine, Qingdao University, Qingdao, China

*For correspondence:
ammszdy@126.com (DZ);
tangli@ncpsb.org.cn (LT)

[4]These authors contributed equally

Competing interest: The authors declare that no competing interests exist.

## eLife Assessment

This **valuable** study delineates the cellular contributions of BMP signaling in liver development and function. The findings are **convincing**, and the study employs state-of-the-art molecular, genetic, and cellular approaches to demonstrate that hepatic stellate cells play a central role in liver health by mediating cell-to-cell crosstalk via the production of specific BMP proteins. This study will be of interest to scientists interested in developmental biology and organ physiology.

**Abstract** The liver is the largest solid organ in the body and is primarily composed of hepatocytes (HCs), endothelial cells (ECs), Kupffer cells (KCs), and hepatic stellate cells (HSCs), which spatially interact and cooperate with each other to maintain liver homeostasis. However, the complexity and molecular mechanisms underlying the crosstalk between these different cell types remain to be revealed. Here, we generated mice with conditional deletion of *Gdf2* (also known as *Bmp9*) and *Bmp10* in different liver cell types and demonstrated that HSCs were the major source of GDF2 and BMP10 in the liver. Using transgenic ALK1 (receptor for GDF2 and BMP10) reporter mice, we found that ALK1 is expressed on KCs and ECs other than HCs and HSCs, and GDF2 and BMP10 secreted by HSCs promote the differentiation of KCs and ECs and maintain their identity. *Pdgfb* expression was significantly upregulated in KCs and ECs after *Gdf2* and *Bmp10* deletion, ultimately leading to HSCs activation and liver fibrosis. ECs express several angiocrine factors, such as BMP2, BMP6, Wnt2, and Rspo3, to regulate HC iron metabolism and metabolic zonation. We found that these angiocrine factors were significantly decreased in ECs from *Gdf2/Bmp10*[HSC-KO] mice, which further resulted in liver iron overload and disruption of HC zonation. In summary, we demonstrated that HSCs play a central role in mediating liver cell-cell crosstalk via the production of GDF2 and BMP10, highlighting the important role of intercellular interaction in organ development and homeostasis.

## Introduction

Almost all organs in vertebrates consist of different cell types that undertake their individual functions to maintain organ homeostasis (*Regev et al., 2017*), such as vascular endothelial cells supplying oxygen and nutrients to the organ (*Wong et al., 2017*) and immune cells, especially tissue-resident macrophages, constantly clearing cell debris and monitoring tissue microenvironments to prevent pathogen infection (*Okabe and Medzhitov, 2016*). Usually, these different cell types originate from

distinct progenitors and differentiate into specialized cell types via the expression of lineage-specific transcription factors during development (*Young, 2011*). Meanwhile, crosstalk between these individual cells within one organ occurs and promotes them to continue to specialize (*Pancheva et al., 2022*; *Armingol et al., 2021*), which is needed for the maintenance of the organotypic phenotypes of these cells. However, the cell-cell communication within organs is largely unknown.

The liver is composed of four major cell types: hepatocytes (HCs), Kupffer cells (KCs), endothelial cells (ECs), and hepatic stellate cells (HSCs). The liver exemplifies the importance of cellular interactions in maintaining organ integrity (*Monga, 2014*; *Ding et al., 2016*; *Azimifar et al., 2014*). For example, secretion of Wnt signaling and BMP2/6 by ECs functions on HCs to control hepatic zonation and iron metabolism (*Planas-Paz et al., 2016*; *Yang et al., 2014*; *Canali et al., 2017*; *Koch et al., 2017*), respectively. Loss of these signals results in liver dysfunction. Recent advances in RNA-sequencing (RNA-seq) technology and algorithms have enabled a better understanding of the cellular composition and potential cellular interactions of the liver (*Efremova et al., 2020*). However, the speculative cues still need experimental validation, and more importantly, it is still unknown what consequences happen if specific cell-cell interactions are interrupted.

HSCs reside in the space of Disse and directly interact with HCs, ECs, and KCs (*Bonnardel et al., 2019*). HSCs are well known for their pathological role in liver fibrosis, as HSC activation is a critical step in the development of chronic liver disease (*Tan et al., 2021*). However, in addition to the storage of retinoids, the physiological role of HSCs in the normal liver remains unclear. ALK1, encoded by the activin A receptor-like type 1 (*Acvrl1*) gene, is the receptor of GDF2 (also known as BMP9) and BMP10. Its mutations result in hereditary hemorrhagic telangiectasia (HHT), which causes vascular malformations in multiple organs, including the liver (*Miyazono et al., 2010*). It has been reported that liver sinusoidal EC-specific *Alk1*-deficient mice exhibit severe liver vascular malformations (*Schmid et al., 2023*), which mimic the liver symptoms observed in HHT patients. We and others demonstrated that ALK1 is needed for KC survival and identity, and the ability of KCs from *Alk1*-deficient mice to capture bacteria from the bloodstream was significantly impaired (*Zhao et al., 2022*; *Guilliams et al., 2022*). GDF2 is mainly expressed in the liver. Using antibody staining, GDF2 was reported to be expressed in human HCs (*Bidart et al., 2012*). Recently, it has been reported that GDF2 and BMP10 have been shown to be mainly expressed by murine HSCs (*Guilliams et al., 2022*; *Tillet et al., 2018*). However, no study has systematically examined which liver cell types secrete GDF2 and BMP10 and regulate ECs and KCs via these two paracrine factors. Here, we generated mice with conditional knockout of both *Gdf2 and Bmp10* in different liver cell types and found that HSCs are the functional source of GDF2 and BMP10. Interestingly, conditional deletion of *Gdf2/Bmp10* from HSCs (*Gdf2/Bmp10*[HSC-KO]) not only resulted in loss of KC and EC identity but also affected HC phenotypes in an indirect way, which ultimately resulted in liver dysfunction, suggesting that HSCs play a central role in orchestrating the interactions between different liver cell types under physiological conditions.

## Results

### HSCs are the functional source of GDF2 and BMP10 in the liver

To generate mice lacking *Gdf2* and *Bmp10* in different liver cell types, *Gdf2*[fl/fl] and *Bmp10*[fl/fl] mice were prepared and mated with *Alb*[Cre], *Clec4f*[Cre], *Tek*[Cre], or *Lrat*[Cre] mice, which have been extensively used in the literature to produce HCs, KCs, ECs, and HSC conditional knockout mice (*Canali et al., 2017*; *Zhao et al., 2022*; *Lee et al., 2021*; *Mederacke et al., 2013*; *Sakai et al., 2019*), respectively. Because GDF2 and BMP10 play a redundant role in regulating KCs (*Zhao et al., 2022*), conditional double knockout mice (*Gdf2*[fl/fl]*Bmp10*[fl/fl]*Alb*[Cre], *Gdf2*[fl/fl]*Bmp10*[fl/fl]*Tek*[Cre], *Gdf2*[fl/fl]*Bmp10*[fl/fl]*Clec4f*[Cre] and *Gdf2*[fl/fl]*Bmp10*[fl/fl]*Lrat*[Cre]) were generated. To systematically investigate which liver cell type produces GDF2 and BMP10, we acquired liver tissues from these mice. Quantitative PCR analysis revealed that *Gdf2* and *Bmp10* mRNA expression was significantly reduced in the liver tissues from *Gdf2*[fl/fl]*Bmp10*[fl/fl]*Lrat*[Cre] (*Gdf2/Bmp10*[HSC-KO]) mice compared to other Cre deleter mice (*Figure 1A*). Using Cre-mediated reporter mice, we demonstrated that *Lrat*[Cre] mice can specifically target HSCs (*Figure 1B*, *Figure 1—figure supplement 1*). BMP10 is mostly expressed in the heart, followed by the liver (*Neuhaus et al., 1999*). However, *Bmp10* expression in the heart was not affected in *Gdf2/Bmp10*[HSC-KO] mice (*Figure 1C*). ALK1 is the receptor of GDF2 and BMP10. To investigate which liver cell types express ALK1 and respond to GDF2 and BMP10, we prepared transgenic ALK1 reporter mice, which

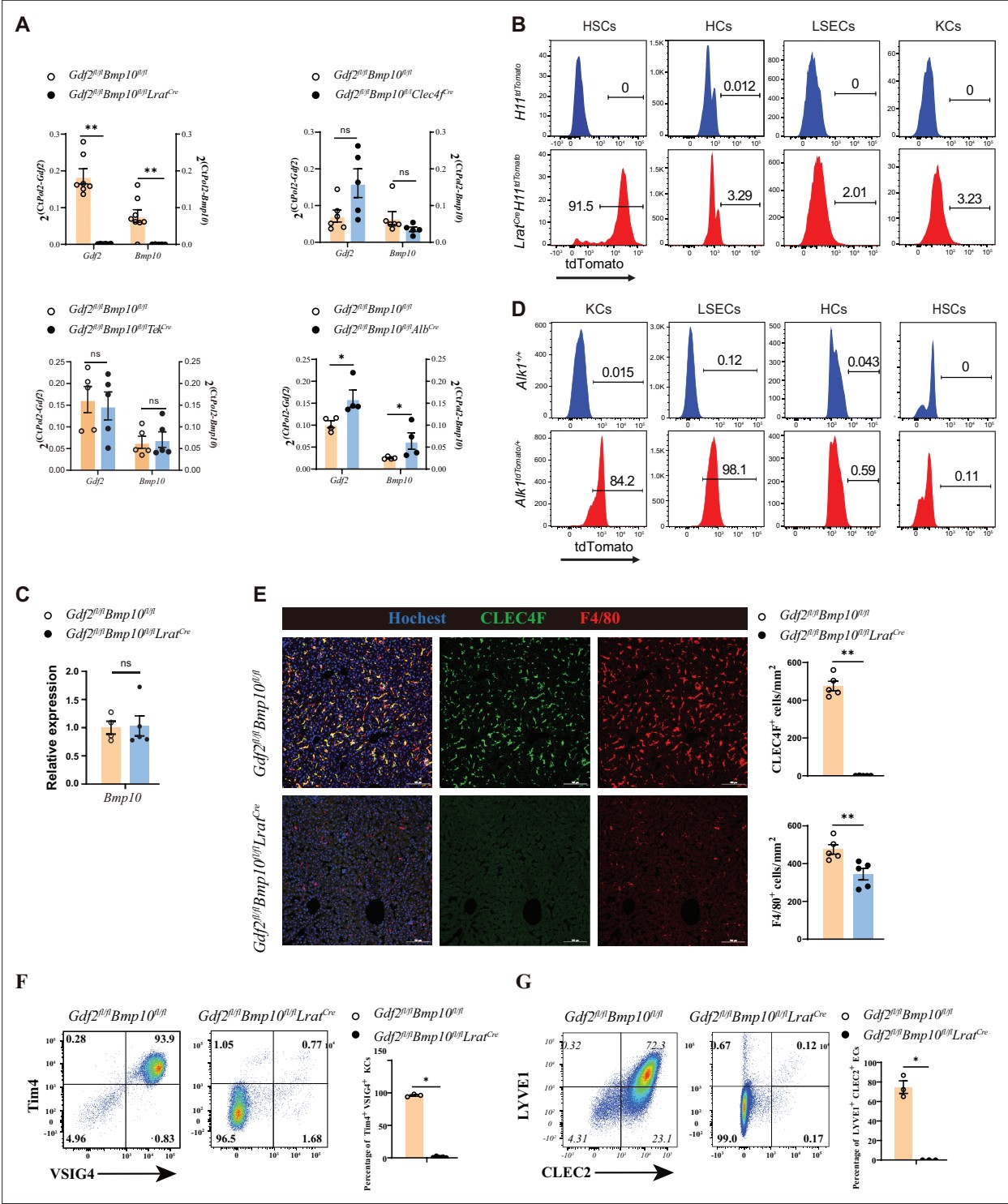

**Figure 1.** Hepatic stellate cells (HSCs) are the major source of GDF2 and BMP10 in the liver. (**A**) Quantitative PCR analysis of *Gdf2* and *Bmp10* expression in the liver from the indicated mice at the age of 8–14 weeks (n=4–6/group). (**B**) Representative flow cytometric expression of tdTomato in HSCs, hepatocytes (HCs), endothelial cells (ECs), and Kupffer cells (KCs) from *Lrat^{Cre}H11^{tdTomato}* mice at the age of 8–9 weeks (n=3/group). (**C**) Quantitative PCR analysis of *Bmp10* expression in the right atrium from *Gdf2^{fl/fl}Bmp10^{fl/fl}Lrat^{Cre}* and their littermate controls at the age of 8–12 weeks (n=4/group). (**D**) Representative flow cytometric expression of tdTomato in HSCs, HCs, ECs, and KCs from *Alk1^{tdTomato}* mice at the age of 8–12 weeks (n=3/group). (**E**) Immunofluorescence images of liver sections from *Gdf2^{fl/fl}Bmp10^{fl/fl}Lrat^{Cre}* and their littermate controls at the age of 3–12 weeks (n=6/group). Scale bars: 100 µm. (**F**) Flow cytometric expression of Tim4 and VSIG4 in KCs (pregated on CD45^{+}Ly6C^{−}F4/80^{+}CD64^{+}) and percentage of Tim4^{+}VSIG4^{+} KCs from *Gdf2^{fl/fl}Bmp10^{fl/fl}Lrat^{Cre}* (n=5) and their littermate controls (n=3) at the age of 8–12 weeks. (**G**) Flow cytometric expression of LYVE1 and CLEC2 in

*Figure 1 continued on next page*

*Figure 1 continued*

ECs (pregated on CD45$^+$CD31$^-$) and percentage of LYVE1$^+$CLEC2$^+$ ECs from *Gdf2$^{fl/fl}$Bmp10$^{fl/fl}$Lrat$^{Cre}$* (n=3) and their littermate controls (n=3) at the age of 8–12 weeks. Results represent the mean ± SEM. *p<0.05, **p<0.01, ***p<0.001, and ****p<0.0001, by Mann-Whitney test (**A, C, E–G**).

The online version of this article includes the following source data and figure supplement(s) for figure 1:

**Source data 1.** Numerical data of *Figure 1A, C and E-G*.

**Figure supplement 1.** Gating strategies.

**Figure supplement 2.** The phenotypes of Kupffer cells (KCs) were not affected in the liver from *Gdf2$^{fl/fl}$Bmp10$^{fl/fl}$Alb$^{Cre}$*, *Gdf2$^{fl/fl}$Bmp10$^{fl/fl}$Tek$^{Cre}$*, and *Gdf2$^{fl/fl}$Bmp10$^{fl/fl}$Clec4f$^{Cre}$* mice.

**Figure supplement 3.** GDF2 can compensate the role of BMP10 in the liver.

**Figure supplement 3—source data 1.** Numerical data of *Figure 1—figure supplement 3A*.

can faithfully reflect the expression of ALK1. We found that ALK1 was expressed on ECs and KCs but not HCs and HSCs (*Figure 1D*), suggesting that GDF2 and BMP10 can directly function on ECs and KCs via ALK1. Accordingly, we found that surface markers of liver F4/80$^+$ macrophages from *Gdf2/Bmp10$^{HSC-KO}$* mice, including CLEC4F, Tim4, and VSIG4, were almost not expressed (*Figure 1E and F*). In contrast, these markers were not affected in the liver F4/80$^+$ macrophages from *Gdf2$^{fl/fl}$Bmp10$^{fl/fl}$Alb$^{Cre}$*, *Gdf2$^{fl/fl}$Bmp10$^{fl/fl}$Tek$^{Cre}$*, and *Gdf2$^{fl/fl}$Bmp10$^{fl/fl}$Clec4f$^{Cre}$* mice (*Figure 1—figure supplement 2*). LYVE1 and CLEC2, the surface markers of CD31$^+$ hepatic ECs, were also reduced in the livers of *Gdf2/Bmp10$^{HSC-KO}$* mice (*Figure 1G*). In addition, we also determined the phenotypes of KCs and ECs from *Bmp10$^{fl/fl}$Lrat$^{Cre}$* mice to exclude the possibility that the altered phenotypes observed in *Gdf2$^{fl/fl}$Bmp10$^{fl/fl}$Lrat$^{Cre}$* mice were due to Cre-mediated cytotoxicity (*Janbandhu et al., 2014*) and found that these cells were not affected (*Figure 1—figure supplement 3*). Collectively, these results suggest that HSCs are the functional source of GDF2 and BMP10 in the liver to regulate KCs and ECs.

## The differentiation of liver macrophages from *Gdf2/Bmp10$^{HSC-KO}$* mice is blocked in a status of monocyte-derived macrophages

To acquire more information about transcriptome changes in liver macrophages in the *Gdf2/Bmp10*-deficient liver microenvironment, we performed bulk RNA-seq analysis on sorted liver macrophages from *Gdf2/Bmp10$^{HSC-KO}$* mice compared with their controls. Principal component analysis (PCA) suggested that liver macrophages from *Gdf2/Bmp10$^{HSC-KO}$* mice were different to that of their controls (*Figure 2A*). We found that 2424 genes were differentially expressed (FC>2, p-adj<0.05), among which 1313 genes were upregulated and 1111 genes were downregulated in liver macrophages from *Gdf2/Bmp10$^{HSC-KO}$* mice compared to their control mice (*Figure 2B*). From these differentially expressed genes (DEGs), we found that genes associated with the identity of KCs (*Scott et al., 2018*), such as *Fabp7*, *Cd5l*, *Cdh5*, and *Clec4f*, were significantly downregulated in liver macrophages from *Gdf2/Bmp10$^{HSC-KO}$* mice (*Figure 2C*). *Id1* and *Id3*, target genes of ALK1 signaling and important regulators of KC differentiation, were also significantly reduced in liver macrophages from *Gdf2/Bmp10$^{HSC-KO}$* mice (*Figure 2C*). We also observed downregulation of embryonic KC marker *Timd4* (encoding Tim4) expression and upregulation of monocytic *Cx3cr1* and *Ccr2* expression (*Bonnardel et al., 2019*; *Scott et al., 2016*) in liver macrophages from *Gdf2/Bmp10$^{HSC-KO}$* mice (*Figure 2D*), suggesting that embryonic KCs were lost and replaced by monocyte-derived macrophages (MoMs)/KCs, as a reduced number of liver macrophages were observed in *Gdf2/Bmp10$^{HSC-KO}$* mice (*Figure 2E*).

Upon KC loss, blood monocytes are recruited into the liver, immediately differentiate into F4/80$^+$ MoMs and then gradually acquire the expression of CLEC2, CLEC4F, and VSIG4 to become monocyte-derived KCs (MoKCs) with time (*Bonnardel et al., 2019*; *Scott et al., 2018*). CLEC2, encoded by the *Clec1b* gene, was thought to be one of the earliest markers gained by MoKCs and can be used to discriminate MoKCs from MoMs (*Tran et al., 2020*). In addition, TREML4 expression is immediately upregulated upon blood monocyte entry into the liver (*Sakai et al., 2019*). Interestingly, we found that nearly all liver macrophages from *Gdf2/Bmp10$^{HSC-KO}$* mice did not express CLEC2 and TREML4, but liver macrophages from *Alk1$^{fl/fl}$Vav1$^{Cre}$* and *Smad4$^{fl/fl}$Vav1$^{Cre}$* mice did (*Figure 2F*), suggesting that the differentiation of liver macrophages from *Gdf2/Bmp10$^{HSC-KO}$* mice is completely blocked in a status of MoMs, and the differentiation defect in liver macrophages from *Gdf2/Bmp10$^{HSC-KO}$* mice is more pronounced than that from *Alk1$^{fl/fl}$Vav1$^{Cre}$* and *Smad4$^{fl/fl}$Vav1$^{Cre}$* mice. Our previous report has shown that *Alk1* and *Smad4* were efficiently deleted in KCs from *Alk1$^{fl/fl}$Vav1$^{Cre}$* and *Smad4$^{fl/fl}$Vav1$^{Cre}$* mice,

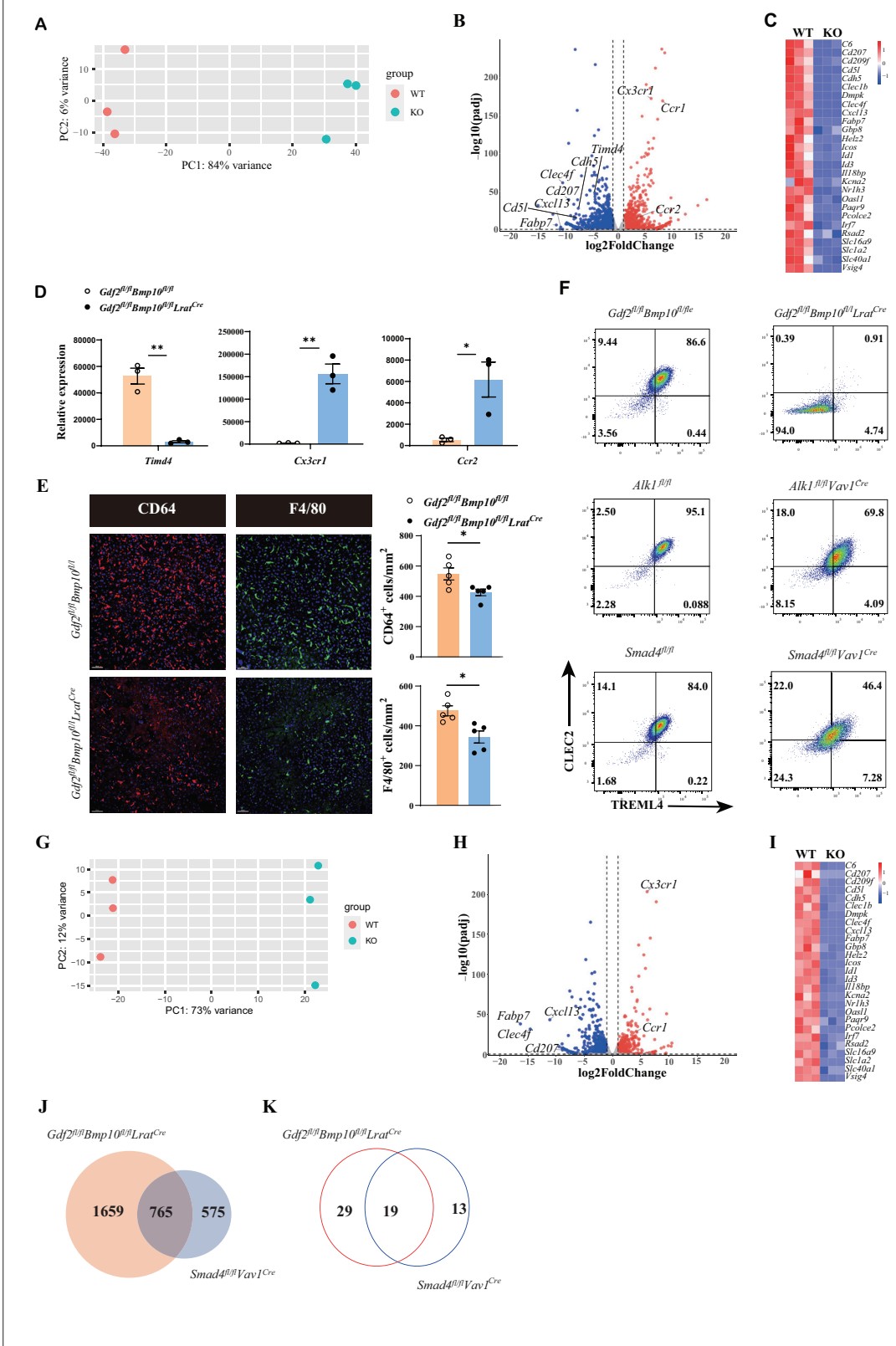

**Figure 2.** The differentiation of liver macrophages was inhibited in *Gdf2^fl/fl Bmp10^fl/fl Lrat^Cre* mice. (**A, B**) Principal component analysis (PCA) and volcano plot of the RNA-sequencing (RNA-seq) data of sorted CD45+Ly6C⁻F4/80+CD64+ liver macrophages from *Gdf2^fl/fl Bmp10^fl/fl Lrat^Cre* and control mice at the age of 8–10 weeks (n=3/group). Genes upregulated and downregulated are shown in red and blue, respectively (fold change [FC]>2,

*Figure 2 continued on next page*

*Figure 2 continued*

adjusted p [p-adj]<0.05). (**C**) Heatmap showing signature genes expressed differentially in liver macrophages from *Gdf2^fl/fl^Bmp10^fl/fl^Lrat^Cre^* and control mice. (**D**) Expression counts of indicated genes in liver macrophages from *Gdf2^fl/fl^Bmp10^fl/fl^Lrat^Cre^* and control mice. \*\*\*p-adj<0.001. (**E**) Immunofluorescence images of F4/80^+^ and CD64^+^ liver macrophages in sections from *Gdf2^fl/fl^Bmp10^fl/fl^Lrat^Cre^* mice and their controls at the age of 12 weeks (n=5/group). Liver macrophages number was measured (right). Scale bars: 50µm. (**F**) Representative flow cytometric expression of CLEC2 and TREML4 in liver macrophages from the indicated mice at the age of 8–12 weeks (n=3–4/group). (**G, H**) PCA and volcano plot of the RNA-seq data of sorted liver macrophages from *Smad4^fl/fl^Vav1^Cre^* and control mice at the age of 8–10 weeks (n=3/group). Genes upregulated and downregulated are shown in red and blue, respectively (fold change [FC]>2, adjusted p [p-adj]<0.05). (**I**) Heatmap showing signature genes expressed differentially in liver macrophages from *Smad4^fl/fl^Vav1^Cre^* and control mice. (**J, K**) Venn diagram showing differentially expressed (DE) genes (**J**) and transcription factors (**K**) specific to liver macrophages of *Gdf2^fl/fl^Bmp10^fl/fl^Lrat^Cre^*, *Smad4*-deficient liver macrophages or shared between both mac populations. Results represent the mean ± SEM. \*p<0.05, by Mann-Whitney test (**E**).

The online version of this article includes the following source data and figure supplement(s) for figure 2:

**Source data 1.** Numerical data of *Figure 2D and E*.

**Figure supplement 1.** RXRα is required for the differentiation from blood monocytes to monocyte-derived Kupffer cells (MoKCs).

**Figure supplement 1—source data 1.** Numerical data of *Figure 2—figure supplement 1C*.

respectively (*Zhao et al., 2022*). To understand this discrepancy, we performed bulk RNA-seq analysis on sorted liver macrophages from *Smad4^fl/fl^Vav1^Cre^* mice, as Smad4 functions as a common Smad needed for transcriptional regulation in response to ALK signaling. In addition to ALK1, GDF2 and BMP10 also bind ALK2 and ALK6, respectively, with low affinity to transduce signals (*Desroches-Castan et al., 2022*). PCA suggested that *Smad4*-deficient KCs were distinguishable from *Smad4*-sufficient KCs (*Figure 2G*). Analysis of the DEGs between *Smad4*-sufficient and *Smad4*-deficient KCs identified 1340 DEGs, including many KC signature genes (*Figure 2H and I*). Comparison of the difference between the DEGs in liver macrophages from *Gdf2/Bmp10*^HSC-KO^ and *Smad4^fl/fl^Vav1^Cre^* mice revealed that 1659 of 2424 DEGs in liver macrophages from *Gdf2/Bmp10*^HSC-KO^ mice did not overlap with DEGs in MoKCs from *Smad4^fl/fl^Vav1^Cre^* mice (*Figure 2J*), which is consistent with the phenomenon described above, indicating that in addition to Smad4, alterations in other molecular mechanisms may affect the phenotypes of liver macrophages in *Gdf2/Bmp10*^HSC-KO^ mice.

Transcription factors play an important role in the differentiation of tissue-resident macrophages. We compared the nonoverlapping transcription factors between the DEGs in liver macrophages from *Gdf2/Bmp10*^HSC-KO^ and *Smad4^fl/fl^Vav1^Cre^* mice and found 29 transcription factors only downregulated in liver macrophages from *Gdf2/Bmp10*^HSC-KO^ mice (*Figure 2K* and *Supplementary file 1*). Among these transcription factors, we focused on *Rxra* because it has been reported to regulate the development of many embryonic tissue-resident macrophages, including KCs (*Philpott et al., 2022*), but it is unclear whether RXRa regulates the differentiation of blood monocytes to MoKCs. To test this hypothesis, we prepared *Rxra^fl/fl^Csf1r^Cre^ Clec4f^DTR^* mice, in which DT injection can delete KCs and monocytes recruited into the liver to differentiate into liver macrophages, which can be targeted by the *Csf1r^Cre^* strain to eliminate the *Rxra* gene (*Figure 2—figure supplement 1A*). We found that RXRα deficiency significantly inhibited the differentiation of blood monocytes to MoKCs, as the proportion of CLEC2^+^ and VSIG4^+^ macrophages was significantly reduced in *Rxra*-deficient macrophages compared with their controls (*Figure 2—figure supplement 1B and C*), suggesting that the differentiation defect of liver macrophages from *Gdf2/Bmp10*^HSC-KO^ mice may be partially caused by *Rxra* downregulation.

## ECs from *Gdf2/Bmp10*^HSC-KO^ mice are transdifferentiated into continuous ECs

Next, to understand how ECs were affected in *Gdf2/Bmp10*^HSC-KO^ mice, we performed bulk RNA-seq analysis on sorted CD45^-^CD31^+^ ECs from *Gdf2/Bmp10*^HSC-KO^ mice compared with their controls. We found that 2692 genes were differentially expressed (FC>2, p-adj<0.05), among which 1534 genes were upregulated and 1158 genes were downregulated in ECs from *Gdf2/Bmp10*^HSC-KO^ mice compared to their control mice (*Figure 3A and B*). Both GATA4 and MAF act as master regulators of hepatic sinusoidal differentiation (*Géraud et al., 2017*; *Gómez-Salinero et al., 2022*), and their expression in

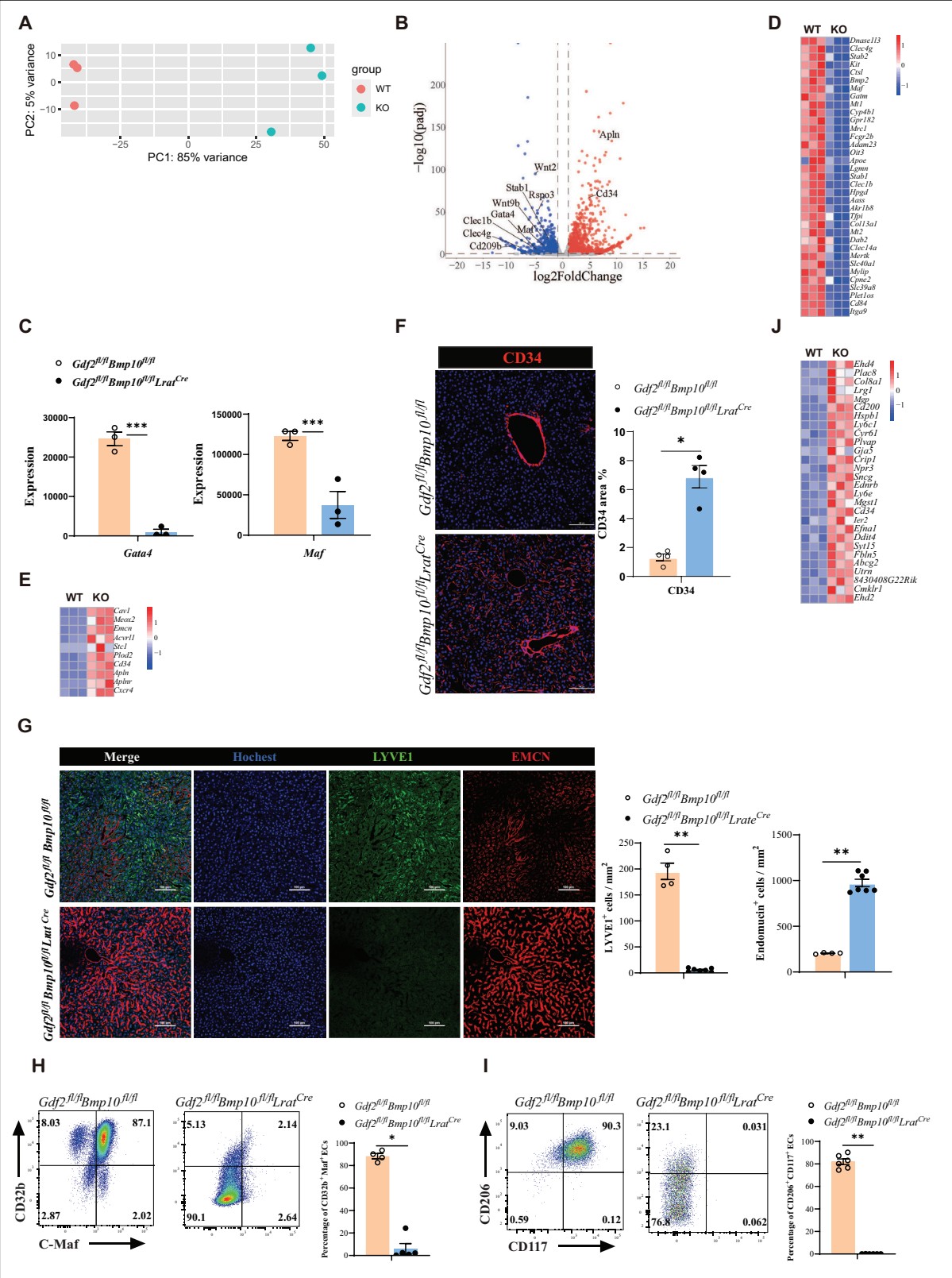

**Figure 3.** Endothelial cells (ECs) from *Gdf2^{fl/fl}Bmp10^{fl/fl}Lrat^{Cre}* mice are transdifferentiated to continuous ECs. (**A, B**) Principal component analysis (PCA) and volcano plot of the RNA-sequencing (RNA-seq) data of sorted hepatic ECs from *Gdf2^{fl/fl}Bmp10^{fl/fl}Lrat^{Cre}* and control mice at the age of 8–10 weeks (n=3/group). Genes upregulated and downregulated are shown in red and blue, respectively (fold change [FC]>2, adjusted p [p-adj]<0.05). (**C**) Expression counts of *Gata4* and *Maf* genes in ECs from *Gdf2^{fl/fl}Bmp10^{fl/fl}Lrat^{Cre}* and control mice. ***p-adj<0.001. (**D, E**) Heatmap showing sinusoidal

*Figure 3 continued on next page*

*Figure 3 continued*

EC-associated genes (**D**) and continuous EC-associated genes (**E**) expressed differentially in hepatic ECs from *Gdf2*<sup>fl/fl</sup>*Bmp10*<sup>fl/fl</sup>*Lrat*<sup>Cre</sup> and control mice.
(**F**) Representative immunofluorescence images in liver sections from *Gdf2*<sup>fl/fl</sup>*Bmp10*<sup>fl/fl</sup>*Lrat*<sup>Cre</sup> and control mice at the age of 12 weeks (n=4/group).
CD34$^+$ area was measured (right). Scale bars: 100 μm. (**G**) Representative immunofluorescence images in liver sections from *Gdf2*<sup>fl/fl</sup>*Bmp10*<sup>fl/fl</sup>*Lrat*<sup>Cre</sup> and
control mice at the age of 8 weeks (n=4–7/group). LYVE1$^+$ and EMCN$^+$ area was measured (right). Scale bars: 100 μm. (**H, I**) Flow cytometric expression
of CD32b, C-Maf, CD206, and CD117 in hepatic ECs from *Gdf2*<sup>fl/fl</sup>*Bmp10*<sup>fl/fl</sup>*Lrat*<sup>Cre</sup> and littermate controls at the age of 8–10 weeks (n=4–6/group).
(**J**) Heatmap showing the indicated genes expressed differentially in hepatic ECs from *Gdf2*<sup>fl/fl</sup>*Bmp10*<sup>fl/fl</sup>*Lrat*<sup>Cre</sup> and control mice. Results represent the
mean ± SEM. *p<0.05 and **p<0.01, by Mann-Whitney test (**F–I**).

The online version of this article includes the following source data for figure 3:

**Source data 1.** Numerical data of *Figure 3C and F-I*.

ECs can be induced by GDF2 in vitro (*Gómez-Salinero et al., 2022*; *Desroches-Castan et al., 2019*).
In our transcriptomic data, *Gata4* and *Maf* expression in ECs from *Gdf2/Bmp10*<sup>HSC-KO</sup> mice were significantly reduced compared with that in their controls (*Figure 3C*). Consistent with the phenotypes of
*Gata4/Maf*-deficient ECs, the expression of sinusoidal EC-specific markers (*Gómez-Salinero et al., 2022*) such as *Stab2*, *Fcgr2b*, and *Mrc1* was significantly downregulated (*Figure 3D*), while the expression of continuous EC-specific markers (*Géraud et al., 2017*) such as *Cd34*, *Emcn*, *Apln*, and *Aplnr*
was upregulated in ECs from *Gdf2/Bmp10*<sup>HSC-KO</sup> mice (*Figure 3E*). Immunostaining revealed obviously
increased expression of CD34 and EMCN, and decreased expression of LYVE1 (*Figure 3F and G*).
Flow cytometry also confirmed the reduced expression of C-Maf (encoded by *Maf*), CD32b (encoded
by *Fcgr2b*), CD117 (encoded by *Kit*), and CD206 (encoded by *Mrc1*) (*Figure 3H and I*). In addition,
genes associated with large arteries (*Gómez-Salinero et al., 2022*) were significantly upregulated
in ECs from *Gdf2/Bmp10*<sup>HSC-KO</sup> mice (*Figure 3J*). These results suggested that hepatic ECs in *Gdf2/
Bmp10*<sup>HSC-KO</sup> mice lost their identity and were transdifferentiated into continuous endothelial and large
arterial cells.

## Liver metabolic zonation and iron metabolism are disrupted in *Gdf2/ Bmp10*<sup>HSC-KO</sup> mice

Angiocrine factors Wnts and BMP2/6 signal were shown to control hepatic metabolic zonation and
iron metabolism (*Planas-Paz et al., 2016*; *Yang et al., 2014*; *Canali et al., 2017*; *Koch et al., 2017*),
respectively. The expression of these genes was significantly reduced in ECs from *Gdf2/Bmp10*<sup>HSC-KO</sup> mice (*Figure 4A*). Prussian blue staining revealed that five out of seven *Gdf2/Bmp10*<sup>HSC-KO</sup> mice
showed mild iron overload in the focal liver (*Figure 4B*). Wnt pathway activation induces the expression of zonated proteins such as glutamine synthetase. Immunostaining confirmed the disappearance
of glutamine synthetase in the livers of *Gdf2/Bmp10*<sup>HSC-KO</sup> mice (*Figure 4C*), suggesting that liver
metabolic zonation was disrupted in the *Gdf2/Bmp10*-deficient liver microenvironment.

To acquire more information regarding how the livers were affected under the condition of GDF2
and BMP10 deficiency, we performed bulk RNA-seq analysis on whole liver samples of *Gdf2/Bmp10*<sup>HSC-KO</sup> mice and their control mice at the age of 12 weeks. We analyzed the expression of Wnt signal
target genes enriched in the pericentral zone and found that most pericentral Wnt signal targets
were significantly downregulated in the livers of *Gdf2/Bmp10*<sup>HSC-KO</sup> mice, including *Glul*, *Oat*, *Cyp2e1*,
*Cyp1a2*, *Cldn2*, *Axin2*, *Lect2*, *Lgr5*, *Tbx3*, and *Rnf43* (*Figure 4D*), supporting the fact that the liver
microenvironment in *Gdf2/Bmp10*<sup>HSC-KO</sup> mice lacks Wnt2, Wnt9b, and Rspo3 signaling. In addition,
GDF2 and BMP10 play a redundant role in regulating liver metabolic zonation, as glutamine synthetase was still expressed in the livers of *Gdf2*<sup>fl/fl</sup>*Lrat*<sup>Cre</sup> and *Bmp10*<sup>fl/fl</sup>*Lrat*<sup>Cre</sup> mice (*Figure 4E and F*).

## Liver fibrosis occurs in *Gdf2/Bmp10*<sup>HSC-KO</sup> mice

Liver weight and the liver/body weight ratio were significantly reduced in *Gdf2/Bmp10*<sup>HSC-KO</sup> mice at
the age of 12 weeks compared with control mice (*Figure 5A*), suggesting that liver development is
impaired. Picrosirius red (PSR) staining revealed increased collagen deposition in the livers of *Gdf2/
Bmp10*<sup>HSC-KO</sup> mice at the age of 12 weeks (*Figure 5B*), suggesting that these mice presented liver
fibrosis. This phenotype was also seen in the livers of double-knockout mice for *Gdf2* (constitutive)
and *Bmp10* (*Bmp10*<sup>fl/fl</sup>*Rosa26*<sup>CreERT2</sup>, tamoxifen inducible) (*Bouvard et al., 2022*). PDGFB is a profibrotic cytokine that results in HSC activation and liver fibrosis. In the transcriptomic data, we found
that *Pdgfb* was significantly upregulated in the liver tissue, ECs, and liver macrophages of *Gdf2/

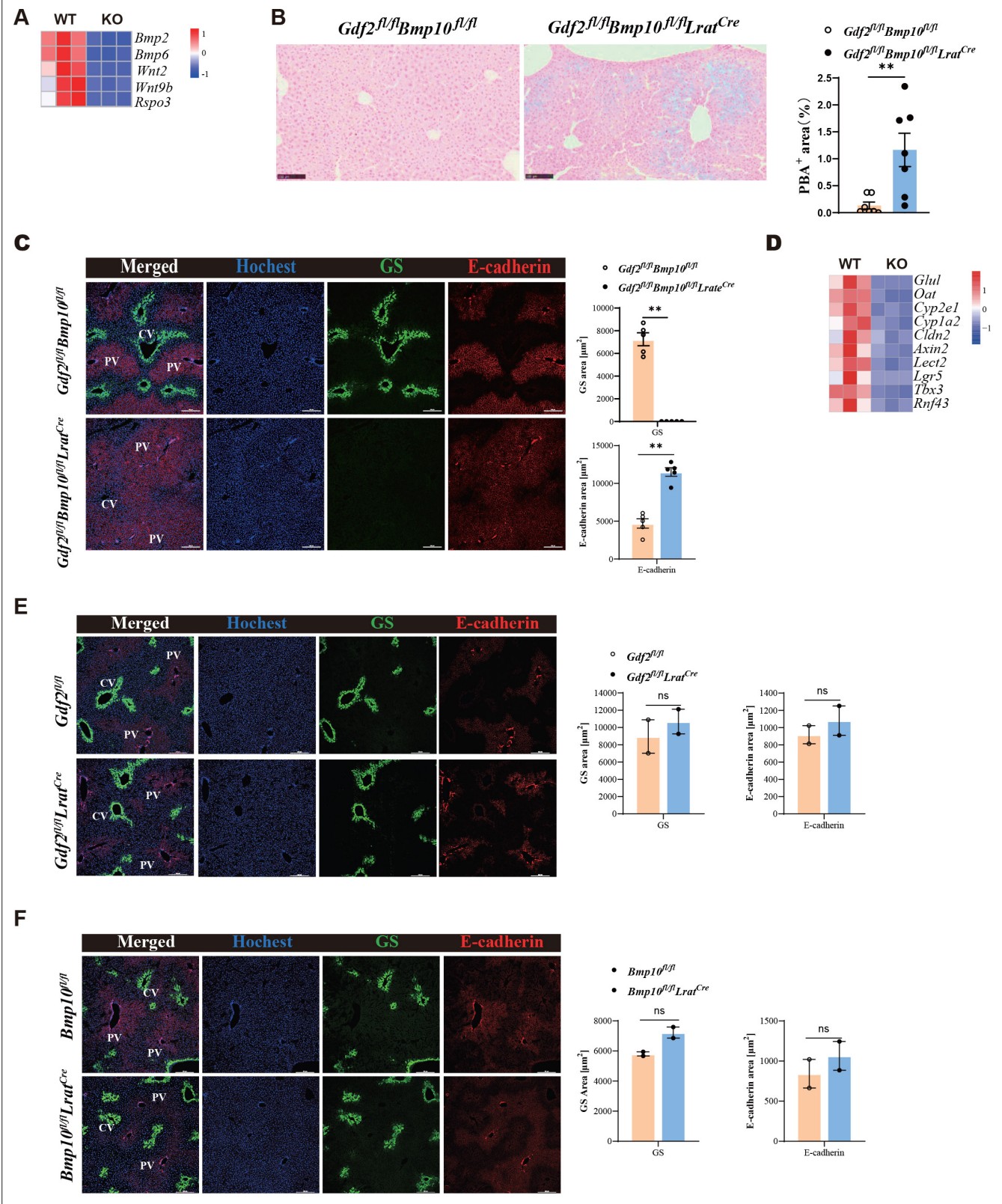

**Figure 4.** Liver metabolic zonation and iron metabolism are disrupted in *Gdf2^{fl/fl}Bmp10^{fl/fl}Lrat^{Cre}* mice. (**A**) Heatmap showing the indicated genes expressed differentially in hepatic endothelial cells (ECs) from *Gdf2^{fl/fl}Bmp10^{fl/fl}Lrat^{Cre}* and control mice. (**B**) Prussian blue in livers from *Gdf2^{fl/fl}Bmp10^{fl/fl}Lrat^{Cre}* mice and their controls at the age of 12 weeks (n=7/group). (**C**) Immunofluorescence images in liver sections from *Gdf2^{fl/fl}Bmp10^{fl/fl}Lrat^{Cre}* mice and their controls at the age of 12 weeks (n=5/group). Scale bars: 200 μm. (**D**) Heatmap showing the indicated genes expressed differentially in liver

*Figure 4 continued on next page*

*Figure 4 continued*

tissues from *Gdf2^fl/fl^Bmp10^fl/fl^Lrat^Cre^* and control mice at the age of 12 weeks. (**E, F**) Immunofluorescence images in liver sections from *Gdf2^fl/fl^Lrat^Cre^* (**E**) and *Bmp10^fl/fl^Lrat^Cre^* (**F**) mice and their controls (n=2/group) at the age of 13–21 weeks (n=2/group). Scale bars: 200 µm. Results represent the mean ± SEM. **p<0.01, by Mann-Whitney test (**B, C, E, F**).

The online version of this article includes the following source data for figure 4:

**Source data 1.** Numerical data of *Figure 4B, C, E and F*.

---

*Bmp10*^HSC-KO^ mice (***Figure 5C***). Immunostaining revealed that the expression of Desmin and type I collagen, two markers of HSC activation, was upregulated in the liver tissues of *Gdf2/Bmp10*^HSC-KO^ mice (***Figure 5D***), suggesting that HSC activity was increased. Endothelial GATA4 deficiency leads to sinusoidal capillarization (***Winkler et al., 2021***). In line with this report, the expression of collagen IV, a marker of EC capillarization, was significantly increased in the livers of *Gdf2/Bmp10*^HSC-KO^ mice, as revealed by immunostaining (***Figure 5E***) and the transcriptomic data of ECs (***Figure 5F***). These results suggested that liver fibrosis in *Gdf2/Bmp10*^HSC-KO^ mice was due to EC capillarization and HSC activation.

## Discussion

In this study, we demonstrated that HSCs convey GDF2 and BMP10 signals to KCs and ECs to maintain their identity. Loss of GDF2 and BMP10 in the liver microenvironment resulted in liver macrophage differentiation defects and a shift in hepatic EC differentiation toward continuous ECs and large arterial cells. Under this condition, physiological crosstalk between ECs and HCs was disrupted, which ultimately altered the whole liver microenvironment and resulted in liver dysfunction.

Recently, we and others have demonstrated that ALK1 is essential for KC differentiation, self-renewal, identity, and endocytic ability (***Zhao et al., 2022***; ***Guilliams et al., 2022***). Here, we undertook systematic genetic manipulation to scrutinize the functional source of GDF2 and BMP10 in the liver. We found that instead of ECs, KCs, and HCs, HSCs are the predominant source of GDF2 and BMP10 critical for KC homeostasis. GDF2 is mainly expressed in the liver, whereas BMP10 is mostly expressed in the heart and weakly expressed in the liver. We found that BMP10 expression in the heart was not affected in *Gdf2/Bmp10*^HSC-KO^ mice, suggesting that the BMP10 signal needed for KCs and ECs was provided by the local liver rather than circulating BMP10 secreted by the heart. Indeed, ALK1 is weakly expressed in KCs and ECs, as the fluorescence signal emitted by KCs and LSECs from *Alk1^td-Tomato^* reporter mice was weak when detected by flow cytometry, suggesting that the concentration of circulating BMP10 may not be enough to activate ALK1 signaling on hepatic ECs and KCs to maintain their phenotypes.

Recently, Schmid et al. used *Stab2^Cre^* mice to delete *Alk1* in liver sinusoidal ECs to model liver HHT (***Schmid et al., 2023***). They found that upon deletion of *Alk1*, liver sinusoidal ECs lost their identity and were transdifferentiated into large arterial cells. In addition, angiocrine Wnt signaling, such as Wnt2, Wnt9b, and Rspondin3, was impaired in *Alk1*-deficient liver sinusoidal ECs, leading to the disruption of liver metabolic zonation (***Schmid et al., 2023***). These phenomena were also observed in the *Gdf2/Bmp10*^HSC-KO^ mouse model. However, liver fibrosis observed in HHT patients did not occur in the livers of *Alk1^fl/fl^Stab2^Cre^* mice (***Schmid et al., 2023***). Indeed, our transcriptomic data showed that *Stab2* expression was significantly reduced in hepatic ECs of *Gdf2/Bmp10*^HSC-KO mice^, which was consistent with the data from *Gdf2* KO 129/Ola mice (***Desroches-Castan et al., 2019***), suggesting that ALK1 signaling regulated STAB2 expression. Thus, it is possible that heterozygosity for the *Alk1* deletion in ECs may result in reduced expression of STAB2, and the expression of Cre recombinase driven by the *Stab2* promoter accordingly decreases, so that the *Alk1*-floxed allele in ECs cannot be efficiently deleted by Cre-mediated cyclization, which may explain why liver fibrosis did not occur in *Alk1^fl/fl^Stab2^Cre^* mice. In view of liver fibrosis, *Gdf2/Bmp10*^HSC-KO^ mice may be a better mouse model to mimic the liver pathology observed in HHT patients (***Desroches-Castan et al., 2019***). In addition, the liver macrophage niche is composed of HCs, ECs, and HSCs (***Bonnardel et al., 2019***; ***Sakai et al., 2019***). Thus, the altered phenotypes of HCs and ECs from *Gdf2/Bmp10*^HSC-KO^ mice may be another reason for the differentiation defect of liver macrophages observed in *Gdf2/Bmp10*^HSC-KO^ mice.

*Gata4*-deficient mice spontaneously develop liver fibrosis (***Winkler et al., 2021***). Single-cell RNA-seq analysis of healthy and cirrhotic human liver samples revealed that *Gata4* expression was also

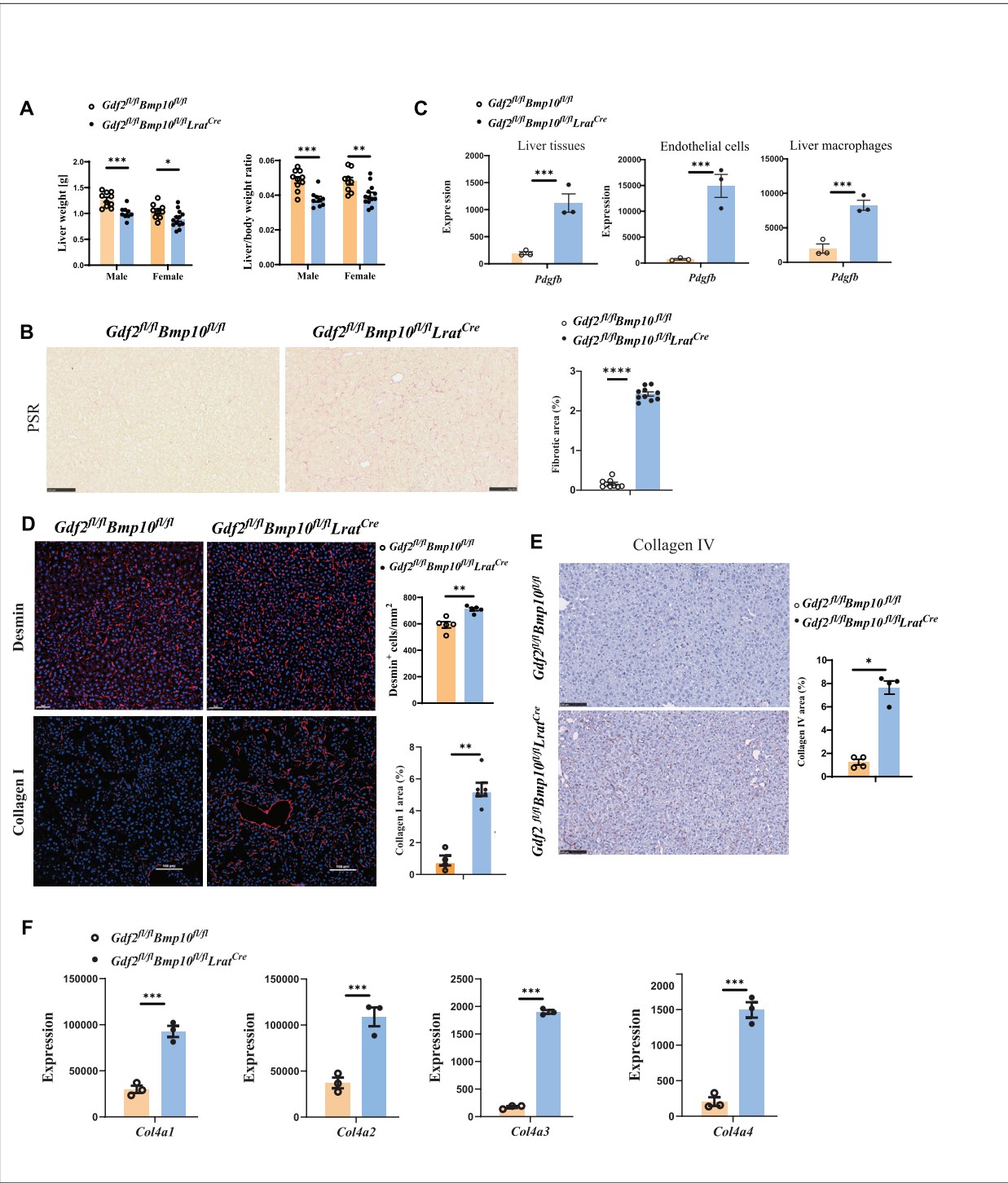

**Figure 5.** Liver fibrosis occurs in *Gdf2^fl/fl^Bmp10^fl/fl^Lrat^Cre^* mice. (**A**) Liver weight and liver to body weight ratio of *Gdf2^fl/fl^Bmp10^fl/fl^Lrat^Cre^* mice and their littermate controls at the age of 12 weeks. (**B**) Picrosirius red (PSR) staining of liver sections from *Gdf2^fl/fl^Bmp10^fl/fl^Lrat^Cre^* mice and their controls at the age of 12 weeks (n=9–10/group). Scale bars: 100 µm. (**C**) Expression counts of *Pdgfb* gene in liver tissues (left), endothelial cells (middle), and liver macrophages (right) from *Gdf2^fl/fl^Bmp10^fl/fl^Lrat^Cre^* mice and their control mice. ***p-adj<0.001. (**D**) Immunofluorescence images of Desmin and collagen I in liver sections in liver tissues from *Gdf2^fl/fl^Bmp10^fl/fl^Lrat^Cre^* and control mice at the age of 12–28 weeks (n=4–6/group). Scale bars: 50µm (upper) and 100µm (lower). (**E**) Collagen IV immunohistochemistry staining of liver sections from *Gdf2^fl/fl^Bmp10^fl/fl^Lrat^Cre^* mice and their controls at the age of 12 weeks (n=4–5/group). Scale bars: 100 µm. (**F**) Expression counts of indicated genes in endothelial cells from *Gdf2^fl/fl^Bmp10^fl/fl^Lrat^Cre^* and control mice. ***p-adj<0.001. Results represent the mean ± SEM. *p<0.05, **p<0.01, ***p<0.001, and ****p<0.0001, by Mann-Whitney test (**A, B, D, E**).

The online version of this article includes the following source data for figure 5:

**Source data 1.** Numerical data of *Figure 5A-F*.

reduced in ECs from cirrhotic human liver samples (*Gómez-Salinero et al., 2022*), suggesting that reduced expression of *Gata4* may contribute to murine and human liver fibrosis. Given that GDF2 and BMP10 regulated the expression of *Gata4* in murine ECs, it is possible that the reduced expression of GATA4 may be due to low expression of GDF2 and BMP10 in liver cirrhosis. However, the expression of *Gdf2* and *Bmp10* was not detected by single-cell RNA-seq of either healthy or cirrhotic human livers (*Ramachandran et al., 2019*; *Aizarani et al., 2019*; *MacParland et al., 2018*; *Lu et al., 2022*) due to the low abundance of *Gdf2* and *Bmp10* mRNA in HSCs. Thus, the correlation between *Gdf2*/*Bmp10* expression and liver fibrosis needs further investigation.

Collectively, we demonstrated that GDF2 and BMP10 sourced from HSCs function on KCs and ECs to maintain their identity, which further directs HCs to wire metabolic function to meet the organ's needs. Upon disruption of this crosstalk, liver dysfunction ensued. This work illustrates the complexity underlying the crosstalk between different liver cell types.

## Materials and methods
### Mice
*Gdf2*^fl/+^ (Stock No: T026617), *Bmp10*^fl/+^ (Stock No: T008480), *Tek*^Cre^ (Stock No: T003764), *Lrat*^Cre^ (Stock No: T006205), and H11 reporter mice (Stock No: T006163) were obtained from GemPharmatech (Nanjing, China). *Alk1*^tdTomato^ mice that an expression cassette encoding tdTomato and a self-cleaving 2A peptide was inserted into the start codon of *Alk1* gene were prepared by MODEL ORGANISMS (Shanghai, China). *Alb*^Cre^ mice (Stock No: 003574) and *Csf1r*^Cre^ (Stock No: 029206) mice were obtained from Jackson Laboratory. *Smad4*^fl/fl^ and *Rosa26*^YFP^ mice were kindly provided by Dr. Xiao Yang (Beijing Institute of Lifeomics). *Alk1*^fl/fl^ mice were kindly provided by Dr. Zhihong Xu (Fudan University, Shanghai). *Vav1*^Cre^ mice were kindly provided by Dr. Bing Liu (Fifth Medical Center of Chinese PLA General Hospital). *Clec4f*^Cre/DTR^ strain mice were established by Nanjing BioMedical Research Institute of Nanjing University (NBRI) using CRISPR/Cas9-mediated genome editing on a C57BL/6J background. Mice and their littermates were used between 8- and 16-week-old unless otherwise specifically indicated. All mice are C57BL/6J background and were maintained at the SPF facilities of the Beijing Institute of Lifeomics. All experimental procedures in mice were approved by the Institutional Animal Care and Use Committee at the Beijing Institute of Lifeomics (IACUC-20210528-18MBL).

### Liver cell suspension preparations, flow cytometry, and antibodies
Briefly, the liver was perfused with approximately 20 ml HBSS containing heparin and EDTA, followed by perfusion with digestion buffer containing collagenase type IV (Sigma) and DNase I (Sigma) for 5 min. The digested livers were then disrupted, and pipetted up and down, and the cell suspension was filtered through a 70 μm cell strainer. Parenchymal cells were separated from nonparenchymal cells by centrifugation at 50×*g* for 5 min for two times. ECs and liver macrophages were further enriched by iodixanol gradient (OptiPrep) as previously described (*Ma et al., 2019*).

Cells were first blocked by anti-CD16/CD32 antibody (2.4G2, TONBO Bioscience, 70-0161-U500) and then stained with other appropriate antibodies at 4°C for 20–30 min. 7AAD (559525, BD) was added to dead cell exclusion. Flow cytometry was conducted using an LSRII Fortessa (BD Biosciences). The acquired data were analyzed with FlowJo software (Tree Star). To sort cell, FACSAria III (BD Biosciences) was used. The following antibodies against mouse proteins were used: anti-CD45-AF700 (30-F11, BioLegend, 103128), anti-CD45-BV421 (30-F11, BioLegend, 103133), anti-Ly6C-PE-Cy7 (HK1.4, eBioscience, 25-5932-80), anti-Ly6G-BV421 (1A8-Ly6g, eBioscience, 48-9668-82), anti-CD115-BV605 (AFS98, BioLegend, 135517), anti-CD64-BV421 (x54-5/7.1, BioLegend, 139309), anti-F4/80-BV711 (BM8, BioLegend, 123147), anti-Tim4-PE (RMT4-54, BioLegend, 130006), anti-VSIG4-APC (NLA14, eBioscience, 17-5752-82), anti-CD31-FITC (390, BioLegend, 102414), anti-CD61-PE-Cy7 (2C9.G2, BioLegend, 104317), anti-CD11b-BV711 (M1/70, BioLegend, 101242), anti-CD11b-APC (M1/70, TONBO Bioscience, 20-0112-U100), anti-LYVE1-BV421 (ALY7, Invitrogen, 48-0443-82), anti-Treml4-PE (16E5, BioLegend, 143303), anti-CLEC2-PE (17D9/CLEC-2, BioLegend, 146104), anti-CLEC2-FITC (17D9, Invitrogen, MA5-28218), anti-CD117-APC (2B8, BioLegend, 105812), anti-CD206-PE (C068C2, BioLegend, 141705), anti-c-Maf-BV421 (sym0F1, Invitrogen, 48-9855-42), CD32b-APC (AT130-2, Invitrogen, 17-0321-82). The gating strategies for KCs, ECs, HSCs, and HCs were shown in *Figure 1— figure supplement 1*.

## Quantitative real-time PCR

Total RNA was isolated with Trizol (Thermo) and cDNA was synthesized with Prime Script RT Reagent Kit (Takara). Quantitative PCR was performed with a SYBR Green PCR kit (Toyobo, Japan) in CFX Connect Real-time PCR detection system (Bio-Rad). The specific qPCR primers used are listed in *Supplementary file 2*.

## RNA-seq analysis

Total RNA of ECs and liver macrophages was isolated with RNeasy plus Mini Kit (QIAGEN). Total RNA of the liver tissues was extracted with Trizol (Thermo). Library preparation was performed with Ultra RNA Library Prep Kit (Illumina) followed by RNA-seq using NovaSeq 6000 (Illumina). All samples passed quality control based on the results of FastQC. Reads were aligned to the mouse genome assembly mm10.GRCm38 using HISAT2. The aligned reads were counted using FeatureCounts. DEGs were determined with DESeq2 program at p-adj (adjusted p-value)<0.05 and logFC (log fold change)>1 or logFC where indicated. All RNA-seq experiments were performed with three independent biological replicates. The heatmap and volcano maps of DEGs were visualized using the 'ggplot2' and 'pheatmap' R package.

## Immunohistochemistry and immunofluorescence

For immunohistochemistry, the livers were fixed in 4% PFA for 12 hr, and then was dehydrated with gradient ethanol and embedded in paraffin. Sections (4 μm) were cut on a Finesse 325 (Thermo Scientific) and adhered to adhesion microscope slides. The paraffin sections were dewaxed and hydrated with xylene and gradient ethanol, and then boiled with antigen retrieval solution to repair antigen. Endogenous peroxidase was blocked with 0.3% hydrogen peroxide for 30 min. Paraffin sections were blocked with phosphate-buffered saline (PBS) containing 3% horse serum for 1 hr, and incubated with avidin and biotin for 15 min, followed by staining with anti-collagen IV antibody (ab19808, Abcam) overnight at 4°C. Sections were incubated with secondary antibody, ABC reagent, and NovaRed peroxidase substrate of Vector-kit (Vector Laboratories) following with the manufacturer's instruction. Images were acquired on a C10730-12 NanoZoomer with a ×20/NA 0.75 objective lens.

For immunofluorescence, the fixed livers were dehydrated in PBS solutions containing 15% and 30% sucrose for 12 hr, and embedded in OCT compound (Sakura Finetek). The samples were cut into 20 μm slices using the Cryotome FSE (Thermo Scientific) and then attached to adhesion microscope slides. Frozen liver sections were then blocked for 2 hr with 1% bovine serum albumin, 0.3 M glycine, and 10% donkey or goat serum before being stained overnight at 4°C with primary antibodies diluted in PBS with 0.2% Tween, secondary antibodies, and Hoechst for 2 hr at room temperature. Sections were mounted with Fluoromount G (YEASEN) and images were acquired on a Nikon A1R plus N-STORM confocal microscope with a ×10/0.45 PLANAPO, ×20/0.75 PLANAPO, or ×40/0.75 PLANFLUOR objective lens. Laser excitation wavelengths of 405 nm, 488 nm, 561 nm, and 647 nm channels were used for sample fluorescence detection. The following antibodies were used: anti-F4/80 (BM8, BioLegend, 123101), anti-CD64 (x54-5/7.1, BioLegend, 139301), anti-E-Cadherin-AF647 (DECMA-1, BioLegend, 147307), anti-CD34 (AM34, Invitrogen, 14-0341-82), anti-Glutamine Synthetase (ab49873, Abcam), anti-Desmin (ab15200, Abcam), anti-type I collagen (ab21286, Abcam), anti-EMCN (eBioV.7C7 (V.7C7), eBioscience, 14-5851-81), anti-LYVE1 (AF2125-SP, R&D Systems), and anti-Clec4F (AF2784, R&D Systems). Fluorescent-conjugated secondary antibodies were purchased from Jackson ImmunoResearch Laboratories, Abcam, and Invitrogen, as donkey anti-rabbit AF594 (711-585-152, Jackson), donkey anti-rat AF488 (ab150153, Abcam), donkey anti-rat AF594 (A21209, Invitrogen), donkey anti-sheep AF647 (A21448, Invitrogen), and donkey anti-goat FITC (A11055, Invitrogen).

Hematoxylin and eosin, PSR, Nile red, and Prussian blue were performed according to standard protocols.

## Image processing and analysis

Quantitative analyses of all images were measured using the Fiji package for ImageJ software (NIH) and the Imaris (Oxford Instruments). For coverage measurements of KCs and HSCs, surface modules in Imaris were used to create CD64 and F4/80 biomarkers. A threshold was determined by the algorithm for background subtraction and manually set to match the cell signals. The number of cells per

unit area was accessed by parameters in Imaris. IHC and PSR staining can be assessed by color deconvolution and the thresholding tool in ImageJ.

## Statistics

Data are presented as mean ± s.e.m. All statistics analyses were performed in GraphPad Prism (GraphPad Software). Statistical significance was assessed by the Mann-Whitney test. $*p<0.05$; $**p<0.01$; $***p<0.001$; ns, not significant. Each symbol represents an individual mouse. Number of animals is indicated as 'n'.

## Acknowledgements

This work was supported by the National Science Fund for Distinguished Young Scholars of China (82225009) and National Natural Science Foundation of China (32270941). We thank Flow Cytometry Facility, Animal Facility (Mr. Chen Qiu), and Imaging Facility of National Center for Protein Sciences, Beijing (NCPSB) for their assistance.

## Additional information

### Funding

| Funder | Grant reference number | Author |
| --- | --- | --- |
| National Science Fund for Distinguished Young Scholars | 82225009 | Li Tang |
| National Natural Science Foundation of China | 32270941 | Dianyuan Zhao |

The funders had no role in study design, data collection and interpretation, or the decision to submit the work for publication.

### Author contributions

Dianyuan Zhao, Conceptualization, Resources, Data curation, Formal analysis, Supervision, Funding acquisition, Investigation, Methodology, Writing - original draft, Project administration, Writing - review and editing; Ziwei Huang, Xiaoyu Li, Resources, Data curation, Formal analysis, Validation, Investigation, Visualization, Methodology; Huan Wang, Data curation, Formal analysis, Validation, Investigation, Visualization, Methodology; Qingwei Hou, Yuyao Wang, Fang Yan, Investigation, Methodology; Wenting Yang, Di Liu, Resources, Investigation; Shaoqiong Yi, Chunguang Han, Yanan Hao, Methodology; Li Tang, Conceptualization, Data curation, Formal analysis, Supervision, Funding acquisition, Investigation, Project administration, Writing - review and editing

### Author ORCIDs

Li Tang (ID) https://orcid.org/0000-0002-6514-9266

### Ethics

All experimental procedures in mice were approved by the Institutional Animal Care and Use Committee at the Beijing Institute of Lifeomics (IACUC-20210528-18MBL).

Reviewer #1 (Public review): https://doi.org/10.7554/eLife.95811.3.sa1
Author response https://doi.org/10.7554/eLife.95811.3.sa2

## Additional files

### Supplementary files

• Supplementary file 1. Transcription factors (TFs) downregulated in liver macrophages from *Gdf2/Bmp10*HSC-KO mice and *Smad4*fl/fl *Vav1*Cre mice.

- Supplementary file 2. Primers used for real-time PCR.
- MDAR checklist

## Data availability

The raw RNA-seq data generated from this study have been deposited in NCBI's Gene Expression Omnibus (GEO) under accession number GSE244429.

The following dataset was generated:

| Author(s) | Year | Dataset title | Dataset URL | Database and Identifier |
|---|---|---|---|---|
| Zhao D, Tang L, Wang H | 2023 | BMP9 and BMP10 coordinate liver cellular crosstalk to maintain liver health | https://www.ncbi.nlm.nih.gov/geo/query/acc.cgi?acc=GSE244429 | NCBI Gene Expression Omnibus, GSE244429 |

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
