## [Editor Report · eLife Assessment]

This **valuable** study delineates the cellular contributions of BMP signaling in liver development and function. The findings are **convincing**, and the study employs state-of-the-art molecular, genetic, and cellular approaches to demonstrate that hepatic stellate cells play a central role in liver health by mediating cell-to-cell crosstalk via the production of specific BMP proteins. This study will be of interest to scientists interested in developmental biology and organ physiology.

---

## [Referee Report · Reviewer #1 (Public review)]

Summary:

The aim of the present work is to evaluate the role of BMP9 and BMP10 in liver by depleting Bmp9 and Bmp10 from the main liver cell types (endothelial cells (EC), hepatic stellate cells (HSC), Kupffer cells (KC) and hepatocytes (H)) using cell-specific cre recombinases. They show that HSCs are the main source of BMP9 and BMP10 in the liver. Using transgenic ALK1 reporter mice, they show that ALK1, the high affinity type 1 receptor for BMP9 and BMP10, is expressed on KC and EC. They have also performed bulk RNAseq analyses on whole liver, and cell-sorted EC and KC, and showed that loss of Bmp9 and Bmp10 decreased KC signature and that KC are replaced by monocyte-derived macrophages. EC derived from these Bmp9fl/flBmp10fl/flLratCre mice also lost their identity and transdifferentiated into continuous ECs. Liver iron metabolism and metabolic zonation were also affected in these mice. In conclusion, this work supports that BMP9 and BMP10 produced by HSC play a central role in mediating liver cell-cell crosstalk and liver homeostasis.

Strengths:

This work further supports the role of BMP9 and BMP10 in liver homeostasis. Using a specific HSC-Cre recombinase, the authors show for the first time that it is the BMP9 and BMP10 produced by HSC that play a central role in mediating liver cell-cell crosstalk to maintain a healthy liver. Although the overall message of the key role of BMP9 in liver homeostasis has been described by several groups, the role of hepatic BMP10 has not been studied before. Thus, one of the novelties of this work is to have used liver cell specific Cre recombinase to delete hepatic Bmp9 and Bmp10. The second novelty is the demonstration of the role of BMP9 and BMP10 in KC Differentiation/homeostasis which has already been slightly addressed by this group by knocking out ALK1, the high affinity receptor of BMP9 and BMP10 (Zhao et al. JCI, 2022).

Weaknesses:

This work remains rather descriptive and the molecular mechanisms are barely touched upon and could have been more explored.

---

## [Author Response]

The following is the authors’ response to the original reviews.

**Reviewer #1 (Recommendations For The Authors):**
Materials and Methods section:Cell gating and FACS sorting strategies need to be explained. There is no figure legend of supplementary figure 4 which is supposed to explain the gating strategy. Please detail the strategy for each cell types.

Thank you for your suggestion. We have given a detailed description about the gating and FACS sorting strategies for different liver cell types in supplementary figure 1. In addition, flow cytometry plots of CD45+Ly6C-CD64+F4/80+ KCs from *Bmp9fl/flBmp10fl/flLrat Cre* mouse were also presented in supplementary figure 1.

The genetic background of the different mouse strains and the age of the mice should be noted on each figure.

All the mice used in our study are C57BL/6 background (method section). The age of the mice has been described on each figure.

The Mann Whitney test instead of the two-tailed student's t-test should be used for the different statistical analyses. Why are the expression counts statically analyzed by 2-tailed Student's t test as they were already identified as DE in RNAseq statistical analysis?

Thank you for your suggestion. Statical methods have been corrected in the revised manuscript.

What is the age of the mice and how many are used for each bulk RNAseq?

This information has been added on the corresponding figure legends.

Figure 1:Figure 1a and c: The qPCR data would be much more interesting if presented as DDct and not as relative value as we do not see the mRNA levels of BMP9 and BMP10 in each Bmp9fl/flBmp10fl/flCre mouse. This would allow to compare the mRNA level of BMP9 versus BMP10. This should be changed in all figures.

The presentation of qPCR data in Figure 1a have been changed, which is allowed to compare the abundance of BMP9 versus BMP10 mRNA. Figure 1c only shows the expression of BMP10, so it is unnecessary to present qPCR data as DDct. In our bulk RNA sequencing data of liver tissues, we found that BMP9 expression counts is higher than that of BMP10, in line with the data from BioGPS.

Figure 1e (IF) and f (FACS), the quantification of these data should be added as shown in Fig2d. What is the difference between Fig1e and Fig2d as they both seem to show the quantification of F4/80 in CTL versus Bmp9fl/flBmp10fl/flLratCre mice. Are the cells sorted in Fig1f and 1e and suppl Fig1b? if yes please precise the strategy. If they are not gated how can the authors obtain 93% of KC? The reference Tillet et al., JBC 2018 should be added in the discussion of figure 1 as it is the first description of BMP10 in HSC.

The quantitative data of Figure 1e and 1f have been added in our revised manuscript. Compared with other tissue-resident macrophages, CLEC4F as a KC-specific marker exclusively expressed on KCs. In our previous report (PMID: 34874921), we demonstrated that BMP9/10-ALK1 signal induced the expression of CLEC4F. The data shown in Figure 1e repeated this phenotype that upon loss of BMP9/10-ALK1 signal, liver macrophages did not express CLEC4F. F4/80 in Figure 1e was used as an internal positive control. Fig2d showed the quantification of F4/80 and CD64, two pan-macrophage markers, which was more accurate to measure the number of liver macrophages, especially given that F4/80 mean fluorescence intensity was reduced in liver macrophages of *Bmp9fl/flBmp10fl/flLrat Cre* mice. Cells in Fig1f, 1e and suppl Fig1b were not sorted and the flow cytometry plots of these cells were pre-gated on live CD45+Ly6C-CD64+F4/80+ liver macrophages. The reference Tillet et al., JBC 2018 has been added in our revised manuscript.

Supplementary 4 should have a detailed figure legend and should appear before gating experiments. What cell subtype is used for each cell type gating. Please add the exact references of all the antibodies used and if they are fluorescently labeled antibodies. Why is the number of lymphocytes noted and how is it calculated? The gating strategy for the Bmp9fl/flBmp10fl/flLratCre mice should also be showed as the number of FA4/80+ and Tim4+ cells are decreased.

A detailed figure legend has been added in original supplementary figure 4 that has been moved to supplementary figure 1 in our revised manuscript. The antibodies used in our study were also used in our previous report (PMID: 34874921) and others (PMID: 31561945; PMID: 26813785). Lymphocytes number on flow cytometry plots will automatically appear when we analyze flow cytometry data, so it does not mean that these selected cells are lymphocytes. To avoid the misunderstanding, these words have been deleted. The gating strategy of CD45+Ly6C-CD64+F4/80+ liver macrophages for the *Bmp9fl/flBmp10fl/flLrat Cre* mice was showed in our revised manuscript (Supplementary Figure 1).

Figure 2:Figure 2a: How many mice were used for bulk RNAseq at what age? Please describe the gating strategy for sorting liver macrophages. The PCA should be shown. The genes represented in Fig2c and cited in the text should be shown on the volcano plot and the heatmap (Timd4, Cdh5, Cd5l). A reference for these KC and monocytic markers should be added in the text.

Control and *Bmp9fl/flBmp10fl/flLrat Cre* mice at the age of 8-10 weeks (n=3/group) were used for bulk RNAseq. This information has been added in Figure 2a legend. The PCA, Timd4 gene and references for these KC and monocytic markers have been shown in our revised manuscript according to your suggestion.

Figure 2b: How are selected the genes represented in the heatmap? The top ones? If it is a KC signature the authors should give a reference for this signature.

These genes were KC signature genes. The reference (PMID: 30076102) has been given in our revised manuscript.

Fig2e: Please explain what is the Vav1 promoter and in which cells it will delete Alk1and Smad4? The authors also need to show that Alk1 and Smad4 are indeed deleted in these mice and in which cell subtype (EC and KC?). This is an important point as the authors conclude that other molecular mechanisms than Smad4 signaling may affect the phenotypes of liver macrophages in Bmp9fl/flBmp10fl/flLratCre.

Cre recombinase of *Vav1Cre* mice is expressed at high levels in hematopoietic stem cells (PMID: 27185381). This strain is widely used to target all hematopoietic cells with a high efficiency (PMID: 24857755). In our previous report (PMID: 34874921), we demonstrated that *Alk1* (Supplemental Figure 6A) and *Smad4* (Supplemental Figure 6G) were efficiently deleted in KCs from *Alk1fl/flVav1Cre* and *Smad4fl/flVav1Cre* mice, respectively. This sentence and reference have been added in our revised manuscript. Homozygous loss of ALK-1 causes embryonically lethality due to aberrant angiogenesis (PMID: 28213819). EC-specific ALK1 knockout in the mouse through deletion of the ALK1 gene from an *Acvrl12loxP* allele with the EC-specific L1-Cre line results in postnatal lethality at P5, and mice exhibiting hemorrhaging in the brain, lung, and gastrointestinal tract (PMID: 19805914). In contrast, *Alk1fl/flVav1Cre* mice generated in our lab did not observe this phenomenon or body weight loss, and still survived at the age of 16 weeks. Thus, we don’t think that ECs can be targeted by *Vav1Cre* strain, at least in our experimental system.

Supl Figure 3 (revised Supl Figure 4): The authors need to explain what cell types are affected by Csf1r-Cre and Clec4fDTR. Have the authors tried to perform a similar experiment in Bmp9fl/flBmp10fl/flLratCre? The legend of the Y axis is not clear, why is CD45+ used in the first bar graph while the other two graphs use F4/80+?

We (PMID: 34874921) and others (PMID: 31587991; PMID: 31561945; PMID: 26813785) have demonstrated that Clec4f specifically expressed on KCs and thus only KCs can be deleted in Clec4fDTR mice after DT injection. CSF1R, also known as macrophage colony-stimulating factor receptor (M-CSFR), is the receptor for the major monocyte/macrophage lineage differentiation factor CSF1. Thus, Csf1r-Cre strain can target monocyte, monocyte-derived macrophage and tissue-resident macrophage including liver, spleen, intestine, heart, kidney, and muscle with a high efficiency (PMID: 29761406). We did not perform a similar experiment in *Bmp9fl/flBmp10fl/flLrat Cre* mice as we have demonstrated that the differentiation of liver macrophages from *Bmp9fl/flBmp10fl/flLrat Cre* mice is inhibited. The other two graphs in Supl Figure 4C were obtained from Supl Figure 4B. Flow cytometry plots in Supl Figure 4B are pre-gated on CD45+Ly6C-CD64+F4/80+ liver macrophages, so it is appropriate to use F4/80+ as an internal control.

Figure 3: Same remarks as in Figure 2. How many mice were used for bulk RNAseq, at what age? The PCA should be shown. How were selected the genes represented in the heatmap? The top ones? A reference should be given for the sinusoidal EC and the continuous EC signatures and large artery signature. Maf and Gata4 should be shown on the volcano plot. A quantification for CD34 IF (Fig3e) as well as for the quantification of the FACS data (Fig 3f) should be added.

Control and *Bmp9fl/flBmp10fl/flLrat Cre* mice at the age of 8-10 weeks (n=3/group) were used for bulk RNAseq. According to your suggestion, other revisions have been made.

Figure 4: A quantification and statistical analysis of Prussian staining area and GS IF should be added not just number of mice which were affected.

A quantification and statistical analysis of Prussian staining area and GS IF has been added.

Minor points:Few spelling mistakes that should be checked.Figure 5a, some bar graphs are missing.

Spelling mistakes and missing bar graphs in Figure 5a have been corrected.

**Reviewer #2 (Recommendations For The Authors):**
The authors should provide some additional information:- Did the single HSC-KO mice for either BMP9 or BMP10 already show partial phenotypes?

We think that under steady state, the phenotype of KCs and ECs, described in our manuscript, in the livers of single HSC-KO mice for either BMP9 or BMP10 was not altered. However, we don’t know whether the role of BMP9 and BMP10 is still redundant in liver diseases or inflammation, which is worth further studying.

- The authors should also stain Endomucin, Lyve1, CD32b on liver tissue to assess endothelial zonation/differentiation in addition to FACS analysis.

In our revised manuscript, we performed immunostaining for Endomucin and Lyve1 and found increased expression of Endomucin and decreased expression of Lyve1 (Figure 3g), suggesting that endothelial zonation/differentiation was disrupt in the liver of *Bmp9fl/flBmp10fl/flLrat Cre* mice compared to their littermates. We did not stain CD32b expression in the liver section as there is no good antibody against mouse CD32b for frozen sections.

- Did the authors assess BMP9/BMP10 effects individually and combined in vitro on KC and EC? Are these likely only direct effects or may they also involve each other (i.e. also cross talk between KC and EC in response to BMP9/10?). This could be assessed in co-culture models.

Using ALK1 reporter mice, we demonstrated that KCs and liver ECs express ALK1.We and others have shown that in vitro stimulation with BMP9/BMP10 can induce the expression of ID1/ID3 and GATA4/Maf in KCs and ECs (PMID: 34874921; PMID: 35364013; PMID: 30964206), respectively. These results suggested that BMP9/BMP10 can directly function on KCs and ECs. Indeed, we are also interested in the crosstalk between KCs and ECs. However, in vitro coculture system can not mimic the interaction between KCs and ECs in the liver as these cells will lose their identity upon their isolation from liver environment. Nevertheless, Bonnardel et al. applied Nichenet bioinformatic analysis to predict that liver ECs provide anchoring site, Notch and CSF1 signal for KCs (PMID: 31561945). Of course, this prediction still needs experimental validation.

- The abstract should be rephrased and more specific focus on BMP related intercellular crosstalk in the liver and its implications for liver health and disease. At the end of the abstract they should also emphasize for which specific fields/topics/diseases these findings are important.

Thank you for your suggestion. The abstract has been rephrased and we hope this abstract could satisfy you.